# Visible-Light Radical–Radical Coupling vs. Radical Addition: Disentangling a Mechanistic Knot



Fernando Aguilar-Galindo [1], Ricardo I. Rodríguez [2], Leonardo Mollari [2], José Alemán [2,3,*] and Sergio Díaz-Tendero [3,4,5,*]

1    Donostia International Physics Center (DIPC), 20018 Donostia-San Sebastián, Spain; fernando.aguilar@dipc.org
2    Organic Chemistry Department, Universidad Autónoma de Madrid, 28049 Madrid, Spain; ricardoi.rodriguez@estudiante.uam.es (R.I.R.); leonardo.mollari@estudiante.uam.es (L.M.)
3    Institute for Advanced Research in Chemical Science (IAdChem), Univiversidad Autónoma de Madrid, 28049 Madrid, Spain
4    Chemistry Department, Universidad Autónoma de Madrid, 28049 Madrid, Spain
5    Condensed Matter Physics Center (IFIMAC), Univiversidad Autónoma de Madrid, 28049 Madrid, Spain
*    Correspondence: jose.aleman@uam.es (J.A.); sergio.diaztendero@uam.es (S.D.-T.); Tel.: +34-914973875 (J.A.); +34-914976831 (S.D.-T.)

**Abstract:** A highly enantioselective protocol has been recently described as allowing the synthesis of five-membered cyclic imines harnessing the selective generation of a β-$Csp^3$-centered radical of acyl heterocyclic derivatives and its subsequent interaction with diverse NH-ketimines. The overall transformation represents a novel cascade process strategy crafted by individual well-known steps; however, the construction of the new C-C bond highlights a crucial knot from a mechanistically perspective. We believe that the full understanding of this enigmatic step may enrich the current literature and expand latent future ideas. Therefore, a detailed mechanistic study of the protocol has been conducted. Here, we provide theoretical insight into the mechanism using quantum chemistry calculations. Two possible pathways have been investigated: (a) imine reduction followed by radical–radical coupling and (b) radical addition followed by product reduction. In addition, investigations to unveil the origin behind the enantioselectivity of the 1-pyrroline derivatives have been conducted as well.

**Keywords:** photocatalysis; density functional theory; pyrroline derivative; radical addition; radical–radical coupling; asymmetric synthesis

## 1. Introduction

It is unarguable that radical-based transformations are currently powerful synthetic tools for enabling unconventional bond constructions and disconnections [1–6]. In particular, deep investigations have been performed for the plausible involvement of photoredox catalysis when seeking to design molecules appreciated in the pharmaceutical area [7–11]. The latter has resulted in a wide library of otherwise unachievable routes for molecular building; in particular, imines have proven to be reliable redox-active scaffolds in several transformations for crafting nitrogen-bearing compounds [12]. For instance, imines may behave as one-electron acceptors during a photoredox catalytic cycle, thus unlocking C-centered radical reactivity and hypothesizing a persistent radical effect (PRE) [13–18] (Scheme 1a). By means of this strategy, several interesting hetero-coupling reactions have been described; however, obtaining ethylene diamine derivatives (via pinacol coupling) consistently remains an undesired side reaction. Conversely, it is found that neutral imines might act as SOMOphiles in the presence of previously formed pseudo-nucleophilic radicals and afterwards the resulting nitrogen-centered species can be reduced (Scheme 1b) [19,20]. Conceptually, these two general ideas are diverse from each other;

nevertheless, several factors may operate to determine the fate of the mechanism as it is properly indicated in most of the reports that summon these principles.

**Scheme 1.** General reactivity of imines in photoredox catalysis: (**a**) Radical-radical coupling, (**b**) Radical addition.

As a mind-refresher [21], it would be useful to explain the logic that was followed for the blueprint crafting (Scheme 2). Using the distance between the two key atoms of the transformation as a design guideline (final desired position of 1,5-N and C), we envisioned a unified two-stage process consisting of: (1) the generation of a $Csp^3$ radical by distal functionalization of acyl heterocycles, followed by its radical coupling or addition to a suitable ketimine, and (2) an intramolecular cyclization giving place to the five-member cyclic imine. Thus, we envisioned that the β-C-H bond dissociation energy (BDE) after enolization may vary in several kcal/mol using a rhodium type catalyst [21]. Therefore, the creation of a chiral environment using a centrochiral complex could command stereocontrol in the process and the crucial carbon-centered radical formation might be feasible by a hydrogen atom transfer (HAT).

**Scheme 2.** The blueprint crafting of our previous work [21].

Delightfully, the obtained results applying the protocol described proved the synthetic utility, although some concerns regarding the mechanistic proposal remained unclear until now. In this study, theoretical insight on the relevant reactions of each mechanism is provided by computing the reaction pathway for each case, locating the rate determining step, and discussing the differences between both. Finally, the crucial aspects that steer the enantioselectivity of the reaction were also investigated.

## 2. Results and Discussion

Herein, Scheme 3 presents the catalytic cycles proposed in Reference [21] for the two plausible mechanisms, labeled as pathway A and pathway B. Both proposals begin with a hydrogen atom transfer (HAT). We have explored the potential energy surface corresponding to this step. It is a process without an enthalpic barrier in which the interaction of DABCO·$^+$ with **I** results in a stable complex, followed by the H abstraction (see Supplementary Information). As a result of the HAT process, a radical species (**II**) is produced. As the HAT reaction does not show an enthalpic barrier, this is not the rate determining step of any of the proposed mechanisms.

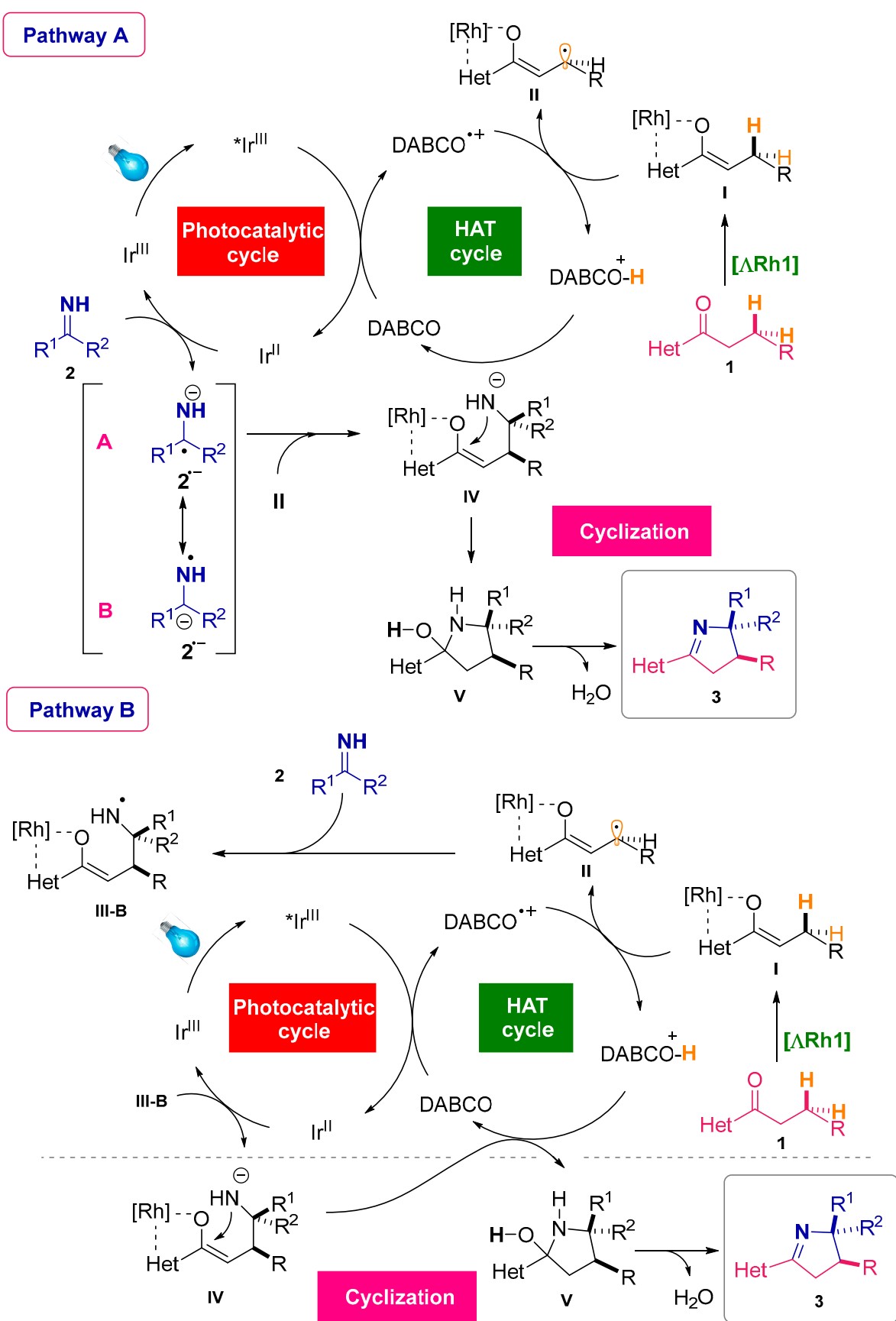

**Scheme 3.** Catalytic cycles proposed in Reference [21] for the two mechanisms (pathway A and pathway B).

In pathway A, the photocatalytic cycle involves the reduction of the imine **2** by the iridium photocatalyst, leading to the radical-anion species **2·⁻**. Subsequently, both radical intermediates interact in a radical–radical coupling, generating species **IV** that evolves towards the final product through a cyclization process. Pathway B shows a different scenario. Firstly, the addition of radical **II** to the imine **2** is suggested to take place, yielding the radical intermediate **III-B**. This compound can be further reduced by a single-electron transfer (SET) process, producing the anion **IV** and regenerating the photocatalyst ground state. Finally, intermediate **IV** evolves into species **3** in the same schedule of pathway A. Based on these assumptions, the radical–radical coupling, radical addition, and photoreduction steps have been separately studied for each proposed mechanism.

Energetic aspects of mechanism A are shown in Figure 1. The first step is a charge transfer from the catalyst (Ir$^{II}$ → Ir$^{III}$) to the imine **2,** delivering **2·⁻**. The thermodynamics of this reaction implies an energy input of about 95 kJ·mol$^{-1}$ when reactants and products are considered at an infinite distance. However, charge transfer processes occur at finite distances and another intermediate situation, where reactants are approaching, has been consistently studied. The obtained results are summarized in Figure 2. As shown, the energy of the first electronic excited states, computed using TD-DFT, are given when the distance between the photocatalyst and the imine ranges from 10 to 30 Å. Most of these states correspond to excitations within orbitals in the photocatalyst; however, the third one is a charge transfer state with an electron promoted from the photocatalyst to the imine. The energy of the charge transfer state as a function of the distance can be fitted to a simple Coulomb interaction with the form:

$$E(r) = A - \frac{B}{r}$$

where *A* and *B* are adjustable parameters. *A* is the excitation energy at an infinite distance. Our fitted value of 71.02 kJ·mol$^{-1}$ misses the redox energy by ~20 kJ·mol$^{-1}$, which is fair considering the simplicity of the model and the fact that the geometries of the imine and catalyst were fixed in the scan. The parameter *B*, with a value of 311.72 Å·kJ·mol$^{-1}$, contains the strength of the interaction, i.e., the charge of both species and effect of the dielectric constant of the solvent. The very good agreement obtained between the simple Coulomb model and the TD-DFT excitation energy values mainly reflects the fact that in this state a cationic (photocatalyst) and an anionic (imine) species are interacting. Charge transfer typically takes place at a distance of ~10 Å [22], corresponding to 40 kJ·mol$^{-1}$ of energy necessary for the process (see bottom-left, Figure 2). Starting from this point, the second step in the mechanism consists of a radical–radical coupling. As reported in Figure 1, the resulting intermediate **IV** is obtained through a barrierless process (stabilized in 27 kJ·mol$^{-1}$). Notice in the figure that a transition state has been located in the potential energy surface; it would correspond to the addition of **II⁻** to **2**.

The results of the second considered pathway, mechanism B, are summarized in Figure 3. In this situation, the radical intermediate **II** directly reacts with the imine via radical addition. This step is energetically quite demanding as it requires ~80 kJ·mol$^{-1}$. However, the radical product **III-B** is easily reduced in the second step of the mechanism which consists of a single electron transfer (SET) from the photocatalyst. In this step, **Ir$^{II}$** is oxidized to **Ir$^{III}$** and the radical **III-B** is stabilized into **IV**.

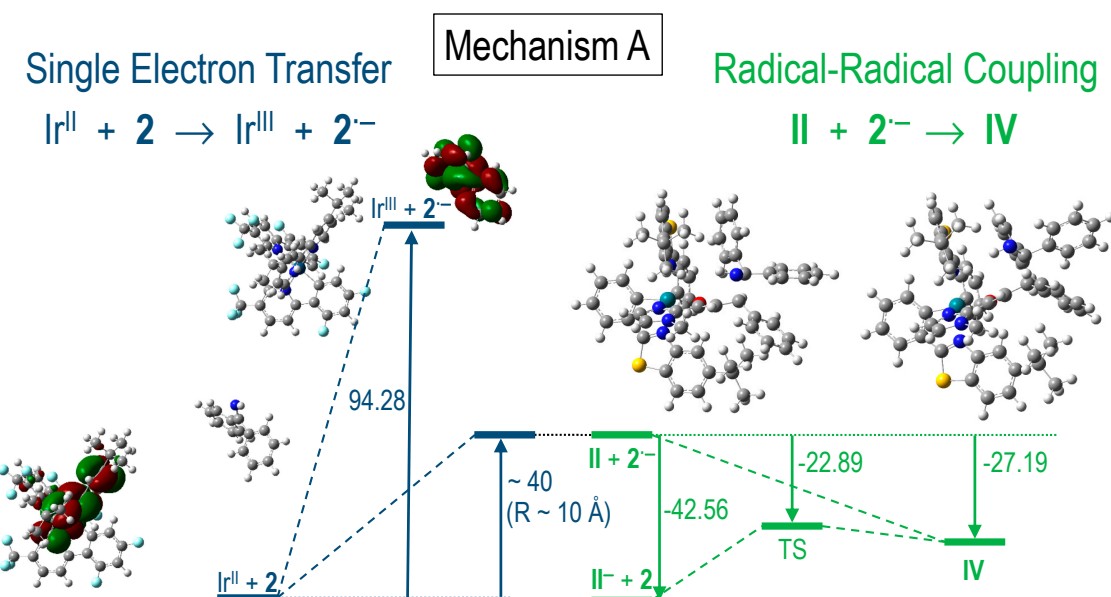

**Figure 1.** Relevant points in the potential energy of the two steps in mechanism A: single electron transfer in the reduction of imine by the photocatalyst (see details in Figure 2) and radical–radical coupling. Relative energies in kJ·mol$^{-1}$ are referred to the separated reactants and have been corrected with the zero-point energy E+ZPE.

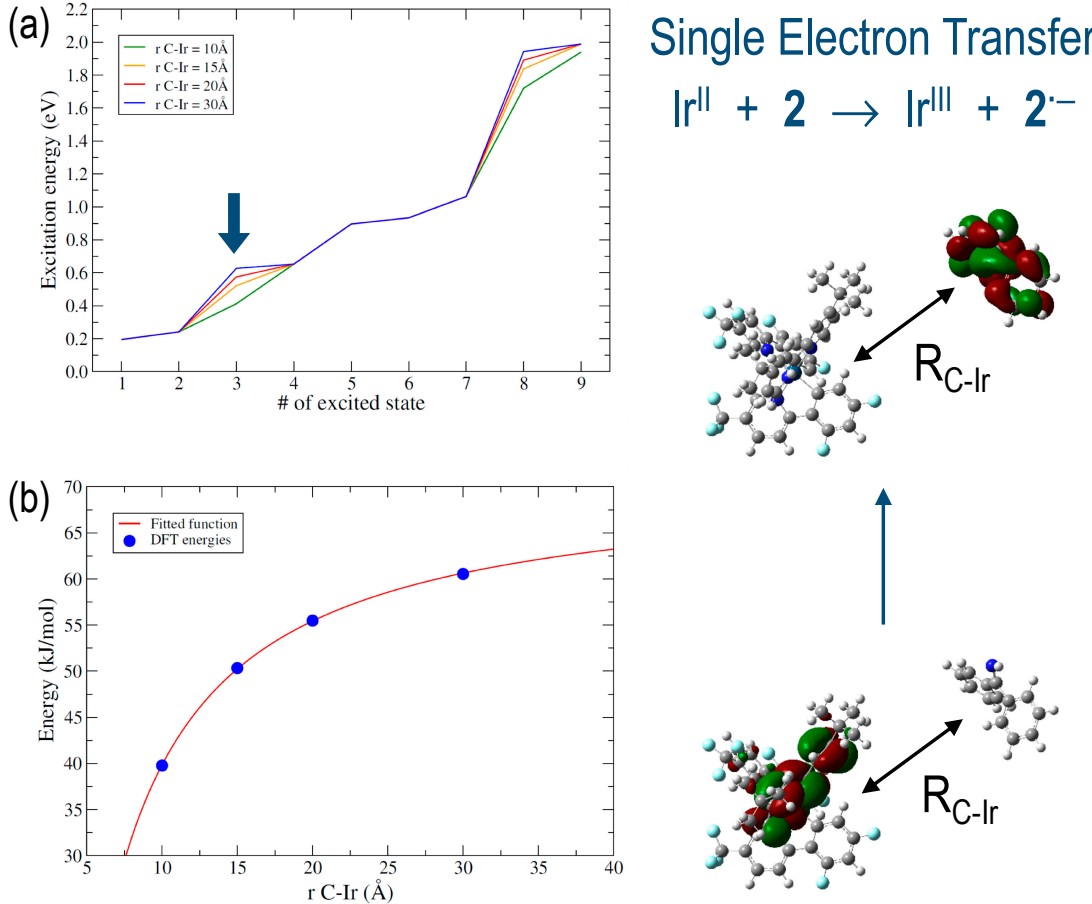

**Figure 2.** (**a**) Excitation energy of the first ten states in the [**Ir$^{II}$** ... **2**] complex at several distances calculated with TD-DFT. Excited state number three (highlighted with a blue arrow) is a charge-transfer state that corresponds to the redox reaction to form (**Ir$^{III}$**)$^+$ and radical **2**$^{·-}$. (**b**) TD-DFT energies of excited state three in panel (**a**) (blue circles) and fitted function E(r) = A − B/r (red line) showing the Coulomb stabilization of the redox reaction as a function of the photocatalysts imine distance.

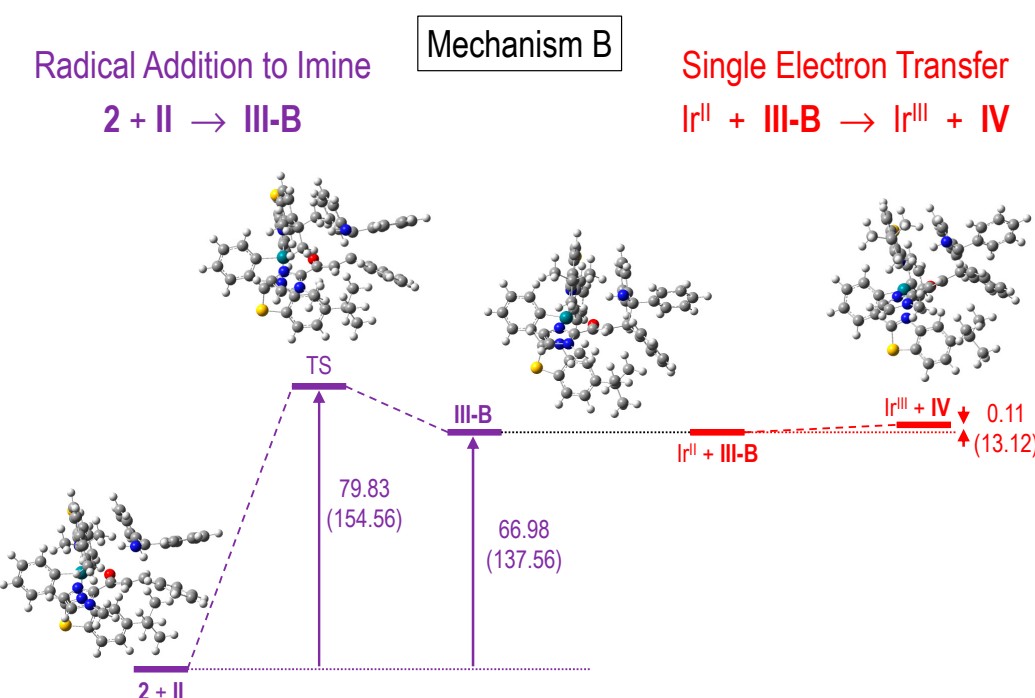

**Figure 3.** Relevant points in the potential energy of the two steps in mechanism B: radical addition to imine and single electron transfer in the reduction of **III-B**. Relative energies in kJ·mol$^{-1}$ are referred to the separated reactants and have been corrected with the zero-point energy E+ZPE.

Thus, as the HAT reaction does not present an enthalpic barrier, the rate determining step in each mechanism is: imine reduction in mechanism A vs. radical addition to imine in mechanism B. Although preliminary results suggested that the first step in mechanism A is a higher-energy process as the charge transfer occurs in a range of ~10 Å, its energy demand is strongly reduced. Therefore, pathway A results the most favorable energetically. This is consistent with the fact that testing the organic photocatalyst 4CzIPN (a neutral organophotocatalys), which is frequently used as surrogate for metallic photocatalysts, [23] did not afford the expected product. By contrast, a different metallic photocatalyst [Ir(ppy)$_2$(bpy)]PF$_6$, a cationic photocatalyst with comparable redox properties to the carbazole derivative, favors the ionic interaction, promoting the reaction.

Lastly, the enantioselectivity of the final products had been investigated. To do this, the first step of both pathways, the hydrogen atom transfer (HAT), was analyzed. Thus, Figure 4a shows the two possible hydrogen abstraction sites of compound **I**; clearly, one of them is much more accessible (blue arrow in the figure) than the other that is sterically hindered. After the HAT process, the radical structure (**II**) adopts a stable planar configuration that helps in delocalizing the unpaired electron. The addition to this compound can take place through two faces. Again, one of them is sterically hindered and only the other is favored, also highlighted with a blue arrow in Figure 4b, explaining the enantioselective synthesis of the 1-pyrroline derivatives. A comparison between the energetic profiles of the addition to both possible options is provided in the Supporting Information.

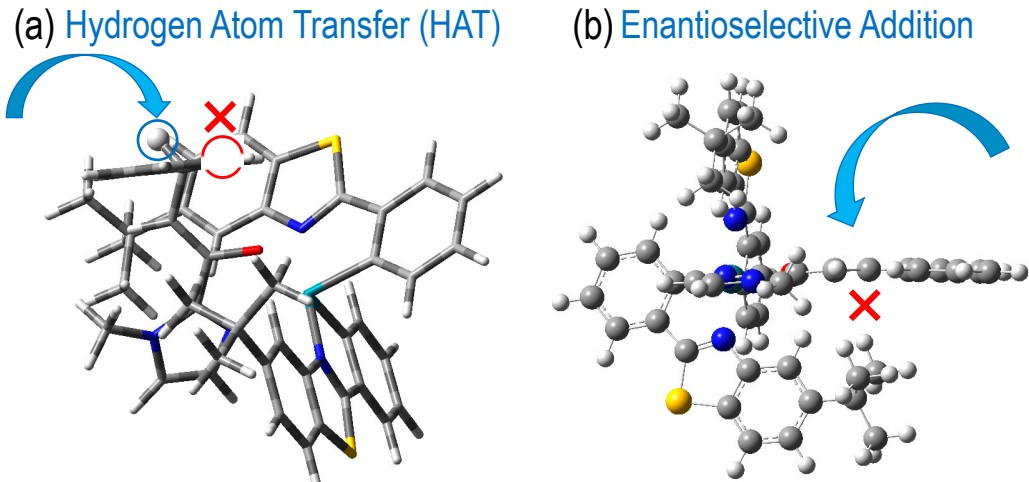

**Figure 4.** (**a**) Tube model of compound **I**. The two hydrogens in the β-position that can be taken in the hydrogen atom transfer (HAT) are in a ball and are highlighted for easier identification. (**b**) Side view of compound **II**, where the sterical hindrance of one of the faces in the addition reaction is clearly shown.

## 3. Computational Details

All calculations were done in the framework of the Density Functional Theory (DFT) with the Gaussian16 [24] package. We have used the well-known B3LYP hybrid functional [25,26] in combination with the Def2SVP split valence basis set [27]. Weak interactions (e.g., dispersion forces) have been included with the Grimme's D3 approach [28]. Solvent (acetonitrile) effects have been introduced with the PCM model [29–31], thus considering the environment and allowing us to describe changes in the electronic density of the chemical species due to the polarizable medium. In the geometry optimization, all the structures have been proved to be a minimum or a first order transition state in the potential energy surface through the analysis of the second derivatives. Electronic excitation energies have been computed within the Time-Dependent Density Functional Theory (TD-DFT) [32,33] with the same combination of the basis and functional. All calculations were performed in absence of symmetry constrains and with the "integral = ultrafine" option.

## 4. Conclusions

In summary, a theoretical study of two possible mechanisms proposed to explain the recent visible-light enantioselective synthesis of 1-pyrroline derivatives [21] was performed. For both plausible mechanisms, the two key steps have been considered. In the first plausible mechanism, mechanism A, a single electron transfer is followed by a radical–radical coupling. In the second plausible mechanism, mechanism B, a radical addition first takes place and then a single electron transfer occurs. An in-depth study of the potential energy surface in these reactions has shown that mechanism A presents the lowest barriers, being the charge transfer the rate determining step and the radical–radical coupling a barrierless process. Reduction in the charge-transfer must occur at a distance between the photocatalyst and the imine around 10 Å. Finally, we also investigated the origin of the enantioselectivity of the pyrroline derivatives. The hydrogen atom transfer of one of the reactants involved in the mechanism conducts to a radical compound with one of its faces sterically unhindered. Hence, addition reactions can take place only through one possible side of the attack.

**Supplementary Materials:** The following are available online at https://www.mdpi.com/article/10.3390/catal11080922/s1, Figure S1: HAT reaction: Energy as a function of the C-H distance. Geometries at the points indicated with arrows are shown, Figure S2: HAT reaction: Energy profile showing critical points. Geometry of the intermediate after hydrogen transfer is shown, Figure S3. Enantioselective addition: Energy profile showing critical points of the favored and non-favored

faces. Relaxed scan of the C-C bond which is being formed in the non-favored face is also given. Energetic profile of the HAT reaction, energetic profile of the addition reaction through the hindered face, optimized geometries of all the compounds studied.

**Author Contributions:** F.A.-G. performed the computations. R.I.R. and L.M. carried out the experiments. J.A. and S.D.-T. conceived the presented idea and wrote the manuscript. All authors discussed the results and contributed to the final manuscript. All authors have read and agreed to the published version of the manuscript.

**Funding:** This research was funded by the Spanish Ministry of Science and Innovation, projects PID2019-110091GB-I00 and RTI2018-095038-B-I00, the European Research Council (ERC-CoG, contract number: 647550), the "Comunidad de Madrid", and the European Structural Funds (S2018/NMT-4367).

**Acknowledgments:** We acknowledge the generous allocation of computer time at the Centro de Computación Científica of the Universidad Autónoma de Madrid (CCC-UAM).

**Conflicts of Interest:** The authors declare no conflict of interest.

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
