# Peer review of "Visible-Light Radical–Radical Coupling vs. Radical Addition: Disentangling a Mechanistic Knot"

_catalysts, doi:10.3390/catal11080922_

Round 1

Reviewer 1 Report

The article by J. Aleman, S. Dìaz-Tendero et al describes a computational study aimed at providing theoretical insight into the mechanism of a previously reported photocatalytic cascade reaction (Angew. Chem. Int. Ed. 2021, 60, 4555–4560) to access enantiomerically enriched substituted chiral enantioenriched 1-pyrroline derivatives. Disappointedly, the supplementary material with the calculation data is missing.

This reviewer sustains that the work would merit publication in Catalysts, but several major issues must be addressed, before acceptance for publication:

  1. Introduction. In my opinion the introduction should be highly improved. First, to understand the main topic of this work, I was forced to download and read ref. 21 (check the year in the ref, should be 2021) as in this text the previously reported transformation was not clearly explained. In fact, commenting Schemes 1 and 2, the authors focused on photoredox modalities to activate imine scaffolds (citing for example the works by McMillan, Ooi and Molander), but only a marginal recall to the author’s previous work has been assessed. I suggest to better introduce ref. 21 as it represents the starting point of the reported study.
  2. About Scheme 3. If intermediates IIIA and IVB represent the same molecule, they should be equally numbered.
  3. About the description of Figures 1 and 3. The description of the potential energy diagrams calculated for mechanisms A and B should be improved. Every calculated intermediate should be numbered and commented in the text.
  4. About Figure 2b. The equation format should be checked.
  5. About Figure 4. Two models were provided to explain the enantioselectivity of the process: regrettably all the computing data are missing (are those minimized structures?). Is it possible to refer to prostereogenic faces?
  6. It's hard to comment on the scientific soundness of a DFT study without all the computing data available. Please let me know if the supplementary material is available.

Author Response

(see attached file)

Reviewer 2 Report

This article presents a theoretical study of the visible-light enantioselective synthesis of 1-pyrroline derivatives. The computational methods used are adequate for the study carried out. The results are clearly presented and discussed. The article deserves publication once some minor points are considered.

  • The size of the figures and schemes should be increased. In some cases it is difficult to read.
  • The authors should unify the designation of TD-DFT or TDDFT along the article.

Acceptable with minor changes.

Author Response

(see attached file)

Reviewer 3 Report

In this manuscript, the authors describe the computational (DFT) study of a photocatalyzed radical cascade process leading to 5-membered cyclic imines.

This study is interesting but it would need a series of improvement before being publishable in Catalysts in my opinion.

Remarks:

- The text contains many grammatical error and should be sent to a native speaker for corrections.

- Some schemes are too small (see for instance Scheme 3).

- Page 5, line 123: the authors claim that “in both mechanisms, the first reaction is the rate determining step”. It would be useful to compute the other steps, especially 1 → II in order to support this statement.

- Page 5, lines 140-142: The analysis of the stereoselectivity is too hand-waving. The authors should compute the two transition states corresponding to the possible addition modes, get their relative energy and investigate the interactions responsible for the enantioinduction.

- Page 6, line 170: they authors write “the radical-radical coupling a barrier-less process”. I guess they mean “no enthalpic barrier”. It should be specified. And my question is then how did they localized a transition state (see Figure 1) if the process does not involve any enthalpic barrier?

Author Response

(see attached file)

Round 2

Reviewer 1 Report

I have read with interest the revised version of this article by J. Aleman, S. Dìaz-Tendero et al dealing with a computational study aimed at providing theoretical insight into the mechanism of a previously reported photocatalytic cascade reaction (Angew. Chem. Int. Ed. 2021, 60, 4555–4560).

In my opinion the work has been improved and it is now in a better shape for publication.

Some minor issues though, should be considered:

About the new Scheme 2: while the “key step” and the “challenge” have been presented in text, no comment has been made about the BDE activation by Rhodium as clearly presented with the green light (a short comment on this should be added for clarity). Also the caption of Scheme 2 is the same of Scheme 3 and should be changed.

In Scheme 3, the radical-anion species 2- should be clearly indicated for clarity.

A general revision of text for minor grammar and punctuation errors should be made: e.g. Scheme instead of scheme and so on…

Overall, this reviewer recommends the publication of this manuscript in Catalysts, provided that the indicated points have been addressed.

Author Response

(please see attached file)

Reviewer 3 Report

The authors have taken most remarks into account and improved their manuscript. I have however two remarks remaining.

- It would have been interesting to compute the two transition states corresponding to the two possible addition modes, get their relative energy and investigate the interactions responsible for the enantioinduction.

- Concerning the first step, the authors write in several places "barrier-less process". This is not true. Since that's an intermolecular process, there is maybe no enthalpic barrier but there is a barrier in terms of free energy. Accordingly, they should write “no enthalpic barrier”.

Author Response

(please see attached file)
